# Expanding the Etiology of Oculo–Auriculo–Vertebral Spectrum: A Novel Interstitial Microdeletion at 1p36

**DOI:** 10.3390/ijms24010036

**Published:** 2022-12-20

**Authors:** Mónica García-Castro, Teresa Martinez-Merino, Nuria Puente, José A. Riancho

**Affiliations:** 1Servicio de Genética, Hospital U. M. Valdecilla, 39008 Santander, Spain; 2Servicio de Medicina Interna, Hospital U. M. Valdecilla, Instituto de Investigación Marqués de Valdecilla (IDIVAL), 39008 Santander, Spain; 3Departamento de Medicina y Psiquiatría, Universidad de Cantabria, 39008 Santander, Spain

**Keywords:** oculo–auriculo–vertebral syndrome, microtia, hemifacial microsomia, chromosomal deletion

## Abstract

The etiology of oculo–auriculo–vertebral spectrum (OAVS) is not well established. About half of patients show a positive family history. The etiology of familiar cases is unclear but appears genetically heterogeneous. This motivated us to report a case of OAVS with microtia, ptosis, facial microsomy, and fusion of vertebral bodies associated with a novel genetic etiology, including a deletion at 1p36.12-13. This case report expands on the genetic etiology of OAVS. Furthermore, it also expands the clinical manifestations of patients with interstitial deletions of the de 1p36.12-13 region.

## 1. Introduction

The abnormal embryonic development of the first and second branchial arches (around the 4th week of gestation) creates various anomalies, mainly of the head and vertebrae, that have been grouped into several syndromes, such as Treacher Collins syndrome, auriculocondylar syndrome, mandibulofacial dysostosis Guion-Almeida type, branchio-oto-renal syndrome, and Goldenhar syndrome [1,2,3,4]. Other disorders with overlapping features include the Miller, Nager, Townes–Brocks, CHARGE, and VACTER syndromes [5]. Some of these disorders have been associated with specific gene variants. For example, 80% of patients with Treacher Collins Syndrome (facial microsomy, usually symmetric, microtia, palatal cleft, eyelid abnormalities) have TCOF1 mutations. Brachio-oto-renal syndrome has been linked to EYA1 or SIX5 gene mutations. However, there is considerable variability and overlap of manifestations. Therefore, the term oculo–auriculo–vertebral spectrum (OAVS) is presently preferred. It is largely equivalent to the conditions known as Goldenhar syndrome or hemifacial microsomia syndrome.

The reported prevalence of OAVS varies between 1/3000 and 1/10,000 births [1]. The phenotype varies from mild to severe. The most common malformations are ear abnormalities (microtia, preauricular tags) with or without hearing loss, facial microsomy (usually unilateral, sometimes bilateral, but almost constantly asymmetric), orofacial clefts, ocular defects (epibulbar dermoids, coloboma, etc.), vertebral abnormalities (fused cervical vertebrae and others), and intellectual disability. Most patients have some degree of asymmetric facial microsomy. Isolated microtia, hemifacial microsomy, mild ear malformations, or hemifacial microsomy plus a family history are usually considered the minimum diagnostic criteria [1]. Patients may have other malformations, particularly of the heart or genitourinary system.

The etiology of OAVS is not well established. About one-third to one-half of cases are sporadic, and one-half show family history [1,6]. Sporadic cases have been associated with acquired disorders during pregnancy, such as maternal diabetes, smoking, placental insufficiency, or drug exposure. The etiology of familiar cases is unclear but appears genetically heterogeneous. Both recessive and dominant forms of inheritance have been described. Thus, much more information is needed to better understand the genetic landscape of these groups of conditions and the genotype–phenotype correlations that help develop deeper scientific knowledge and tailor genetic counseling. This motivated us to report a case of OAVS with a novel genetic etiology.

## 2. Case Presentation

A 33-year-old Caucasian man was sent to our Adult Rare Disease Clinic with suspicion of a malformation syndrome. He was born after an uneventful pregnancy, without known maternal diseases or toxic exposures. Physical and mental development was normal. He had a malformed rudimentary right ear and atresia of the external auditory canal that underwent reconstructive surgery. His parents recalled that the ear was rudimentary, with only the lobule present. He also had been operated on for right palpebral ptosis and used an advancement orthosis for mandibular hypoplasia. His medical history was otherwise unremarkable. He had no siblings and there was no family history of craniofacial abnormalities or other known genetic disorders. 

His height, weight, and cephalic perimeter were 176 cm, 96 kg, and 57 cm, respectively. The right ear was reconstructed. There was residual right ptosis and mild left ptosis, as well as facial asymmetry, with mild left hypoplasia of the face and tongue without orofacial clefts (Figure 1). The skin was normal and there was no joint hypermobility. The rest of the physical exam was normal.

Routine blood chemistries and CBC were normal. Skeletal X-rays showed mild scoliosis and partial fusion of vertebrae C6-C7 and T1 (Figure 1). EKG showed an anterosuperior left-branch block. A cardiac ultrasound revealed no anomalies. An abdominal ultrasound was normal. A DXA scan showed moderately high bone mineral density, with Z-scores (sex and age-adjusted) of +1.5 at the lumbar spine and +1.9 at the hip.

Comparative genomic hybridization (CGH) array was performed using the Sure TAG DNA Labeling kit and Human CGH Microarray kit 4x180K (Nimgenetics, Agilent Technologies, California), following manufacturer’s instructions. Copy number variations (CNVs) were detected with the ADM-2 algorithm implemented in the Agilent Cytogenomic Software v4.0. Only CNVs with at least five consecutive probes (±40 kb) were analyzed. This analysis showed a heterozygous 437.7 kb deletion at 1p36.13 (Chr1.19085207-19522963; GRCh37, hg19) that included the OMIM genes TAS1R2 (taste receptor type 1, member 2), ALDH4A1 (aldehyde dehydrogenase, family 4, subfamily a, member 1), and UBR4 (ubiquitin protein ligase e3 component n-recognin 4), and a few other genes (IFFO2, MIR1290, and MIR4695). The study of both parents was normal (Figure 2). 

Additionally, a clinical exome study of the patient showed three heterozygous variants, namely an incidental pathogenic variant of CFTR (cystic fibrosis transmembrane conductance regulator, located on 7q31) and variants of unknown significance (VOUS) in TNXB (tenascin XB, a member of the tenascin family of extracellular matrix glycoproteins, located on 6p21) and AEBP1 (AE binding protein 1, located on 7p13) genes. The CTFR variant (c.1521-1523 del CTT; p.Phe508del) had been previously associated with cystic fibrosis; thus, the patient was an asymptomatic heterozygous carrier. No cases of cystic fibrosis were reported in the family. The heterozygous TNXB missense variant (p.Gly1265Arg) was not previously reported. TNXB variants have been associated with “TNXB-related type 1 classical-like Ehlers–Danlos syndrome”, a disorder that is similar to classic Ehlers–Danlos, with generalized joint hypermobility, hyperextensible skin, and easy bruising. A recessive mode of inheritance was reported, although some heterozygous relatives showed mild manifestations [7,8]. AEBP1 encodes a member of the carboxypeptidase A protein family. The encoded protein may function as a transcriptional repressor and appears to influence the differentiation of adipocytes and smooth muscle cells, as well as the organization and remodeling of the extracellular matrix [9,10]. The AEBP1 variant (p.Glu352del) affects a poorly conserved residue and has been reported in 0.003% of Europeans. Other AEBP1 variants have also been associated with Ehlers–Danlos syndrome, such as classic-like 2, which also possesses a recessive mode of inheritance [9]. The patient did not show joint hypermobility or any other manifestation suggestive of Ehlers–Danlos syndrome. The exome study did not reveal variants in other genes known to be associated with craniofacial abnormalities.

## 3. Discussion

OAVS seems to be heterogeneous regarding both the genetic background and clinical manifestations. Therefore, we can speculate that a better delineation of the clinical–genetic entities will be possible in the future. However, much more information needs to be accumulated to attain that goal. Mutations of several genes have been implicated in case reports of OAVS (Table 1). They include PAX1, EYA3, MYT1, AMIGO2, ZYG11B, VWA1, SF3B2, ZIC3, YPEL1, CRKL, OTX2, and EFTUD2. Most cases were sporadic; an autosomal dominant mode of inheritance was usually suggested in familiar cases [5,11,12]. Additionally, chromosomal abnormalities have been described in a few patients with OAVS. Thus, Guida et al. [13] performed chromosomal microarray analysis in 19 patients with OAVS and identified pathogenic CNVs in two cases and VOUS in seven. Similarly, Spinelli-Silva et al. found pathogenic CNVs in 4 out of 17 cases and VOUS in another four cases. Their own literature review revealed pathogenic CNVs in 9% patients. Previously reported CNVs were located at 1p22, 5q13.2, 5q15, 10p14, 12p13, 14p31, 15q24, 22qter, and 22q11. However, the genetic variants were rarely recurrent. Deletions at the 22q11 region seem to be the most common (Table 2), although they are only present in approximately 5% of patients [14,15]. Thus, in view of these variable and overlapping genetic and clinical data, more information is needed to fully understand the etiologic landscape of OAVS and the possible genotype–phenotype relationships. Thus, here, we present a patient with OAVS and a previously unreported deletion at chromosome 1.

Terminal 1p36 is considered one of the most common terminal deletions, with an incidence of 1 in 5000 live births. The phenotype of the monosomy 1p36 syndrome is variable and includes intellectual disability, craniofacial dysmorphism, growth delay, eye problems, and hypoacusis [26].

The critical region was initially identified as a 6.3 Mb region at 1p36.33-1p36.31. A more proximal deletion syndrome was described at 1p36.23-1p36.22 by Kang et al. The five patients reported by Kang showed various combinations of microcephaly, prominent forehead, ptosis, rotated and enlarged ears, bulbous nose, digital contractures, and cardiac malformations, among others. The smallest region of overlap (SRO) was located at 1p36.22 [27]. Aagard-Nolting et al. recently reported seven patients from five families with an even more proximal deletion at 1p36.13-1p36.1. The most common manifestations were intellectual or learning disability and ptosis, which were both present in five out of seven patients. Other frequent features included epicanthus, thick eyebrows, high palate, protruding chin, misalignment of teeth, and heart malformations [28]. The SRO encompassed 1 Mb at Chr1:19077793-20081292). The genes responsible were unclear.

The present patient carried a de novo deletion that was within the SRO of Aagard-Nolting’s cases. The region included three protein-coding genes: TAS1R, ALDH4A1, and UBR4. TAS1R encodes a taste receptor; ALDH4A1 encodes a mitochondrial matrix NAD(+)-dependent dehydrogenase, which catalyzes the second step of the proline degradation pathway, converting pyrroline-5-carboxylate to glutamate; and ALDH4A1 deficiency causes hyperprolinemia, a recessive disorder presenting with epilepsy and other neurological problems. UBR4 (Ubiquitin protein ligase e3 component n-recognin 4) is highly expressed in nervous tissue and has been associated with ataxia and other neurological phenotypes. Thus, none of those genes seems to be a biologically plausible candidate to explain the patient’s phenotype. A different driver gene is likely involved. In this view, it is interesting to note that two microRNAs are also transcribed from the deleted region (miR1290 and miR4695). miR1290 is contained within intron 1 of the ALDH4A1-coding region, and it seems involved in cell proliferation through the interaction with its target genes, which may include multiple genes, such as FOXA1, IGFBP3, and others related to the Wnt and hypoxia pathways [29]. The Wnt pathway is a major regulator of the differentiation of many cell types, including mesenchymal and skeletal cells [30,31]. miR4695 may also influence the Wnt pathway by targeting TCF4, a nuclear transcription factor involved in β-catenin signaling [32]. This region is far distant from the 1p22.2-1p31.1 deletion reported by Callier in a patient with severe mental disability, narrow ears, facial asymmetry, and clinodactyly who was diagnosed as having Goldenhar syndrome [14].

## 4. Conclusions

This case report expands the genetic etiology of OAVS. Furthermore, it also expands the clinical manifestations of patients with interstitial deletions of de 1p36.12-13 region. More data are needed to define the genotype–phenotype links among patients with microtia and other abnormalities of the first branchial arches.

## Figures and Tables

**Figure 1 ijms-24-00036-f001:**
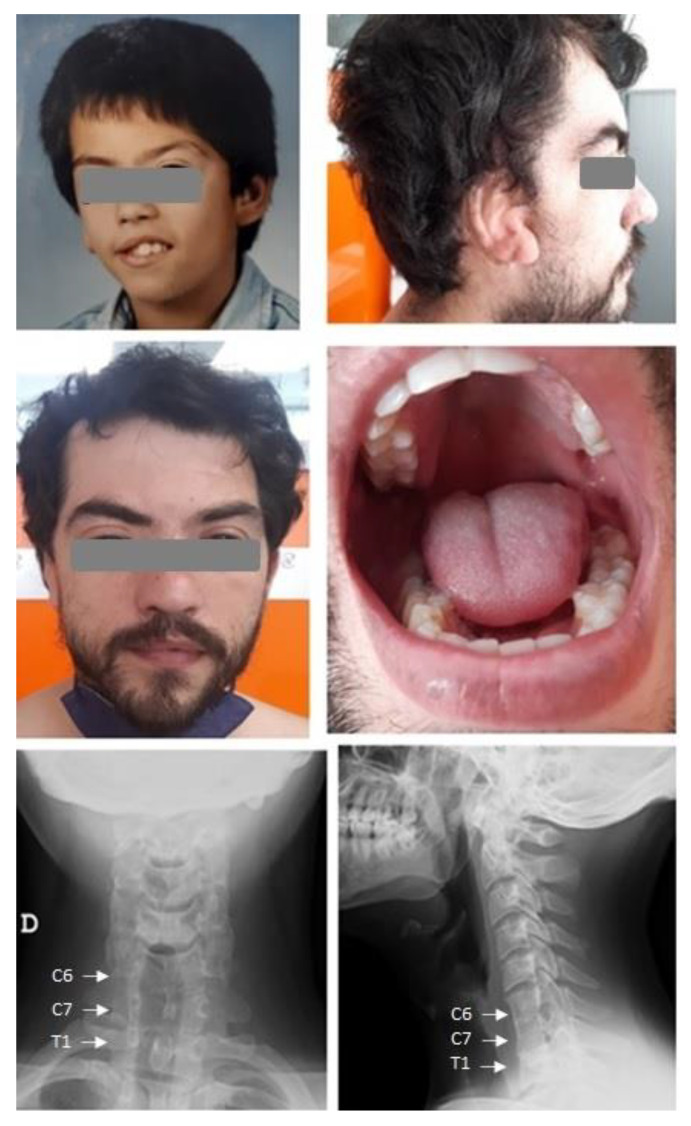
The patient’s pictures show ptosis, thick eyebrows, mild right face hypoplasia, bulbous nose, right hemilingual hypoplasia, and reconstructed ear in childhood (upper left) and adulthood. X-rays show C6-C7-T1 fusion.

**Figure 2 ijms-24-00036-f002:**
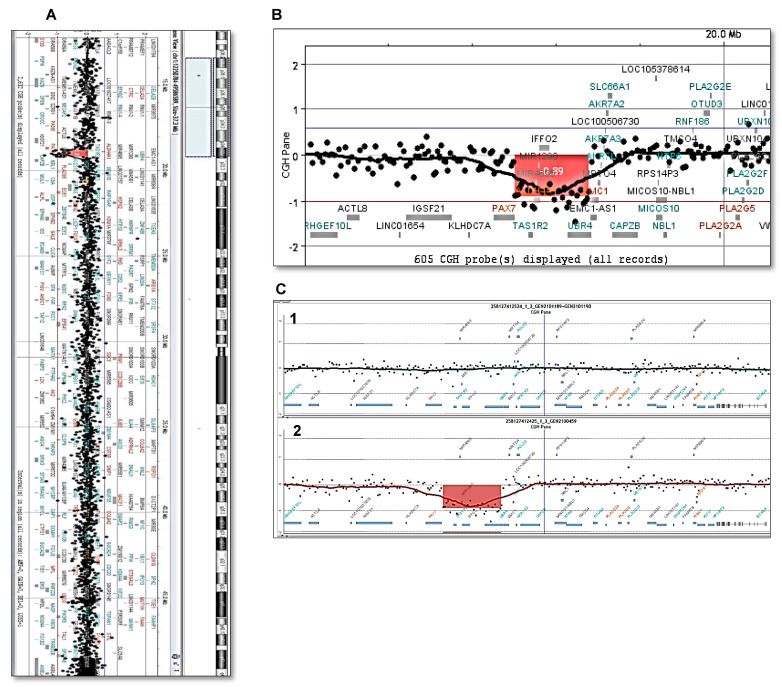
(**A**). CGH array showing an approximately 438 kb deletion in the 1p36.13 region (19085207_19522963) detected in this patient. (**B**). Cytogenomics [GRCh37] showing the TAS1R2, ALDH4A1, and UBR4 genes in the deleted region. (**C**). Parents’ CGH arrays did not show the deletion. The regions of interest in the parents’ DNA (**C1**) (mother’s and father’s DNA 1:1 mix) and the patient’s DNA (**C2**) are shown for comparison.

**Table 1 ijms-24-00036-t001:** Genes with variants associated with OAVS.

Gene Variants	Chromosome	Probands	Ref.
*SF3B2*	11q13.1	7	[5]
*MYT1*	20q13.33	6	[5,16]
*EYA3*	1p35.3	2	[11]
*ZYG11B*	1p32.3	1	[5]
*VWA1*	1p36.33	1	[17]
*ZIC2*	13q32.3	1	[5]
*AMIGO2*	12q13.11	1	[18]
*YPEL1*	22q11.21-22	2	[19]
*CRKL*	22q11.21	1	[19]
*OTX2*	14q22.3	1	[19]
*PAX1*	20p11.22	1	[12]
*EFTUD2*	17q21.31	2	[20,21]

**Table 2 ijms-24-00036-t002:** Some chromosomal abnormalities associated with OAVS. Del, deletion; Dupl, duplication.

Location	Probands	Ref.	Location	Probands	Ref.
1p22-p31 del	1	[15] **	13q34 dupl	1	[15] **
2p12 dupl	1	[15] **	14q22.3 dupl	10	[22]
3q29 dupl	1	[23]	14q23 dup	1	[15] **
4p15.1 dupl	1	[20]	14q31 del	1	[15] **
5p15 del	1	[15] **	15q24 del	1	[15] **
5q13.2 del	1	[1]	15q26.2 del	1	[13] **
5q31.2 dup	1	[13] **	16p13.11 (*)	1	[20]
7q21.11 del	1	[24]	16p13.3 del	1	[15] **
8p22 del (*)	1	[20]	17q11 dupl	1	[15] **
8q13.3 del	1	[15] **	22q11 del/dup	>20	[15,25] **
10p14 dupl	1	[1]	22qter del	1	[1]
10q26.2 del	1	[15] **	Xp22 del	1	[15] **
12p13 del	1	[15] **			

* Additionally, present in a healthy parent. ** These copy number variations (CNVs) were regarded as pathogenic/likely pathogenic; other CNVs of unknown significance were reported in this series.

## Data Availability

Raw data are available from authors by reasonable request.

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
