# Peer review of "Expanding the Etiology of Oculo–Auriculo–Vertebral Spectrum: A Novel Interstitial Microdeletion at 1p36"

_ijms, 2022, doi:10.3390/ijms24010036_

Round 1
Reviewer 1 Report
Oculo-auriculo-vertebral spectrum (OAVS) is a rare developmental disorder. Only a few recurrent CNVs and genes have been identified as causatives and the etiology is unclear in most cases. This manuscript demonstrates an OAVS case and provides valuable genetic analysis results. To improve the paper, I have some suggestions:
1. In Abstract, “oculo-auriculo-vertebral spectrum” should be used instead of “oculo-auriculo-vertebral syndrome” to be consistent with the title and the text. It would be helpful to summarize the phenotypes of the patient here.
2. Figure 1. In the X-rays, I suggest labeling C6, C7, and T1, and adding an arrow to indicate the fusion.
3. Figure 2. The resolution of this figure is low. High-quality images are required to show the genes in the deleted region. What platform was used for CGH arrays?
4. Page 2, Line 74, “The study of both parents was normal (fig 2)”, but in Fig 2C, only one parent’s CGH array result is shown. Is the result of the other parent available?
5. Page 2, Line 74, “Additionally, a clinical exome...”, the result of the exome study is important, so it should be written in another paragraph and more details are needed. What are the variants identified? I suggest listing these variants in a table and showing more information. What are the genotypes of the parents for these variants?
6. Page 2, Line 76, “These genes have been linked to Ehlers-Danlos syndrome“- References should be provided. The functions of these genes should be discussed.
Author Response
- In Abstract, “oculo-auriculo-vertebral spectrum” should be used instead of “oculo-auriculo-vertebral syndrome” to be consistent with the title and the text. It would be helpful to summarize the phenotypes of the patient here.
R. Suggested changes have been done
- Figure 1. In the X-rays, I suggest labeling C6, C7, and T1, and adding an arrow to indicate the fusion.
R. Done
- Figure 2. The resolution of this figure is low. High-quality images are required to show the genes in the deleted region. What platform was used for CGH arrays?
R. The resolution has been enhanced as mucha as possible. Details of the CGH are now given in the text.
- Page 2, Line 74, “The study of both parents was normal (fig 2)”, but in Fig 2C, only one parent’s CGH array result is shown. Is the result of the other parent available?
R. As now stated in the legend, for efficiency reasons, the DNA of both parents was pooled
- Page 2, Line 74, “Additionally, a clinical exome...”, the result of the exome study is important, so it should be written in another paragraph and more details are needed. What are the variants identified? I suggest listing these variants in a table and showing more information. What are the genotypes of the parents for these variants?
R. Much more details about genes and variants in the exome are now given.
- Page 2, Line 76, “These genes have been linked to Ehlers-Danlos syndrome“- References should be provided. The functions of these genes should be discussed.
R. Details and several new references have been included in the revised version
Reviewer 2 Report
I thank the invitation to review the Case Report manuscript entitled “Expanding the etiology of oculo-auriculo-vertebral spectrum: a novel interstitial microdeletion at 1p36”, sent for publication in International Journal of Molecular Sciences.
This manuscript provides the description of a patient with clinical features included in the OAV spectrum and correlated with a “the novo” microdeletion at 1p36.12-13. Some points should be evaluated by the authors:
1. Gene representation should be presented with italics.
2. In Figure 1, there is a picture of an opened mouth but no specific features were represented or described. What does it represent? It is not described in the legend of the figure.
3. Another suggestion would be to include a Table describing the different phenotypes and genetic basis which have been previously correlated to the oculo-auriculo-vertebral spectrum. Example: MIM number, gene, pattern of inheritance, locus, clinical findings, and special features of note.
4. 1p36 deletion syndrome has been widely described previously in the literature with a clinical picture mainly dominated by severe intellectual disability, neurobehavioral changes, seizures, hypotonia, dysphagia, as well as mild dysmorphic features involving mainly the craniofacial segment. The 1p36 deletion syndrome has been previously associated with skeletal, urogenital and cardiac involvement. Why have the authors considered the diagnosis of OAV syndrome and not only the 1p36 deletion syndrome?
Author Response
- Gene representation should be presented with italics.
R. Italics are now used
2. In Figure 1, there is a picture of an opened mouth but no specific features were represented or described. What does it represent? It is not described in the legend of the figure.
R. The hemilingual atrophy is now mentioned in the legend
3. Another suggestion would be to include a Table describing the different phenotypes and genetic basis which have been previously correlated to the oculo-auriculo-vertebral spectrum. Example: MIM number, gene, pattern of inheritance, locus, clinical findings, and special features of note.
R. Although an extensive review of OAVS is out of the scope of this mansucript, we now include two tables with SNVs and CNVs reported in OAVS.
4. 1p36 deletion syndrome has been widely described previously in the literature with a clinical picture mainly dominated by severe intellectual disability, neurobehavioral changes, seizures, hypotonia, dysphagia, as well as mild dysmorphic features involving mainly the craniofacial segment. The 1p36 deletion syndrome has been previously associated with skeletal, urogenital and cardiac involvement. Why have the authors considered the diagnosis of OAV syndrome and not only the 1p36 deletion syndrome?
R. ALthough there may be some overlap between OAVS and 1p36 deletion syndrome, we feel this case lack intellectual dissability and other important features of 1p36 del. On the other hand, features of this case, such as microtia, are more characteristic of OAVS than of typical 1p36.
Round 2
Reviewer 2 Report
The authors have addressed all the changes suggested by the reviewers, and these changes certainly increased the quality of the manuscript, especially in the "Discussion" and "Case Report". Tables 1 and 2 have been perfectly presented by the authors. I congratulate the authors and also highlight the importance of providing the references regarding each gene variant. I have no additional comments or suggestions at this point.